# Experimental H1N1pdm09 infection in pigs mimics human seasonal influenza infections

Theresa Schwaiger[1], Julia Sehl[1,2], Claudia Karte[3], Alexander Schäfer[4], Jane Hühr[4], Thomas C. Mettenleiter[2], Charlotte Schröder[1], Bernd Köllner[4], Reiner Ulrich[1,5], Ulrike Blohm[4]*

1 Department of Experimental Animal Facilities and Biorisk Management, Friedrich-Loeffler-Institut, Greifswald-Insel Riems, Germany, 2 Institute of Molecular Virology and Cell Biology, Friedrich-Loeffler-Institut, Greifswald-Insel Riems, Germany, 3 Institute of Diagnostic Virology, Friedrich-Loeffler-Institut, Greifswald-Insel Riems, Germany, 4 Institute of Immunology, Friedrich-Loeffler-Institut, Greifswald-Insel Riems, Germany, 5 Institute of Veterinary Pathology, Faculty of Veterinary Medicine, University of Leipzig, Leipzig, Germany

* Ulrike.Blohm@fli.de

**Data Availability Statement:** All relevant data are within the manuscript and its Supporting Information files.

## Abstract

Pigs are anatomically, genetically and physiologically comparable to humans and represent a natural host for influenza A virus (IAV) infections. Thus, pigs may represent a relevant biomedical model for human IAV infections. We set out to investigate the systemic as well as the local immune response in pigs upon two subsequent intranasal infections with IAV H1N1pdm09. We detected decreasing numbers of peripheral blood lymphocytes after the first infection. The simultaneous increase in the frequencies of proliferating cells correlated with an increase in infiltrating leukocytes in the lung. Enhanced perforin expression in αβ and γδ T cells in the respiratory tract indicated a cytotoxic T cell response restricted to the route of virus entry such as the nose, the lung and the bronchoalveolar lavage. Simultaneously, increasing frequencies of CD8αα expressing αβ T cells were observed rapidly after the first infection, which may have inhibited uncontrolled inflammation in the respiratory tract. Taking together, the results of this study demonstrate that experimental IAV infection in pigs mimics major characteristics of human seasonal IAV infections.

## Introduction

Influenza A virus (IAV) infections cause low mortality but high morbidity in humans worldwide annually, while pandemics that occur at irregular intervals may have a disastrous impact on global human health [1–3]. Because newly emerging IAV were often of swine origin or arose from reassortments in the pig as mixing vessel [4–7], pigs have increasingly been evaluated as a biomedical model for human influenza [8]. Pigs are anatomically, physiologically and genetically similar to humans indicating a closer mimic of the situation in humans than rodent models. Studies on the immune response in pigs have long been biased by the lack of specific reagents. Especially the variety of antibodies available for the pig is still far lower to that of mice. As a result, the knowledge of the porcine immune response in general is much smaller compared to that of mice. Despite these disadvantages, pigs have decisive advantages as a

**Funding:** This study was funded by Federal Excellence Initiative of Mecklenburg Western Pomerania and European Social Fund (ESF) Grant KoInfekt (ESF_14-BM-A55-00xx_16) to TCM.

**Competing interests:** The authors have declared that no competing interests exist.

model for human IAV infection: they are themselves a natural host for influenza viruses and source of new IAV pathogenic for humans. Thus, understanding the immune response of pigs upon IAV infection and protection by tailored vaccines would assist in the burden on human health by minimizing the emergence of new viruses in swine and zoonotic transmission [4, 5, 9]. Initially, these approaches primarily aimed at identification of antigen preparations that elicit a protective antibody response against surface glycoproteins [10–13], which are effective in controlling IAV infection. However, constant antigenic drift requires an incessant update of the vaccines to match circulating strains. Thus and by expanding the number of reagents for porcine immunological analyses, in the past two decades research increasingly focused on analysis of the immune response against conserved antigenic regions unlikely to change extensively, including cellular immunity. These studies provided evidence for the importance of the cellular immune response in eliminating IAV, also in pigs. These studies have in common that they either only investigated blood-derived cells [14–16] or analyzed the systemic and local immune response only at a very early [17] or late time points of infection [15]. The situation is further complicated by the use of different IAV strains. Recent work recorded for the first time the kinetics of T cell responses and their phenotyping, which supported the previous assumption of the induction of IAV-specific CD4$^+$ and CD8$^+$ T cells [18, 19]. This is in line with several publications reporting IAV-specific CD8$^+$ T cell responses in humans associated with cytolytic [20, 21] as well as memory characteristics [22]. Further, IAV-specific memory T cells were reported to reside in human lungs [23]. Altogether, these porcine studies provided further evidence for similarities of the IAV immune response in pigs and humans [22, 24, 25]. However, it is important to note that the prominent porcine populations of CD4$^+$/CD8$^+$ double positive T cells [26] as well as the high number of peripheral γδ T [27] cells are virtually absent in humans [28, 29], representing a major difference. Besides the numerical difference, the functionality of the two cell populations is comparable in both human and swine. CD4$^+$/CD8$^+$ double positive T cells are mature effector cells with memory characteristics that rapidly mount antigen-specific responses upon antigenic challenge [18, 30]. Besides acting as innate immune cells via pattern recognition receptors and direct killing of infected cells, γδ T cells do play a major role in antigen processing and presentation in human and swine [31, 32]. The two most detailed characterizations of T cell dynamics in influenza-infected pigs were performed by Gerner's group [18, 19]. They reported increased capacity of CD4$^+$/CD8α$^+$ T cells to co-produce IFN-γ, TNF-α and IL-2 as well as the appearance of blood-derived IAV-specific CD8β$^+$ T cells [18]. Further investigations revealed increased frequencies of IFN-γ and/or TNF-α producing, influenza-specific CD4$^+$ and CD8$^+$ T cells in the lungs of infected pigs, resulting in an accumulation of memory cells in the lungs at 6 weeks post infection [19]. Although these porcine IAV studies report a pronounced involvement of the cellular immune response, a global analysis and comparison of immune cells and their functions along the route of entry (nose, lung and lung lymph node) as well as the systemic responses (blood) after infection with H1N1pdm09 were still missing. To ascertain the value of the pig as a reliable model mimicking human IAV infection, we compared the immune response in pigs after a natural IAV infection without inducing major clinical signs, as usually occurs in seasonal human influenza.

## Material and methods

### Ethical statement and study design

All animal experiments were approved by the State Office for Agriculture, Food Safety and Fishery in Mecklenburg-Western Pomerania (LALFF M-V) with reference numbers 7221.3-1-035/17.

**Table 1. Summary of sampling days and animals during study.**

|  | d0 1st infection | d2pi | d4pi | d7pi | d14pi | d21pi 2nd infection | d22pi | d25pi | d31pi |
|---|---|---|---|---|---|---|---|---|---|
| Blood samples | 6 | 6 | 6 | 6 | 6 | 6 | 6 | 6 | 6 |
| Organ samples |  |  | 5 | 5 |  | 5 |  | 5 | 3 |

Blood samples were taken from the same six animals, randomly chosen on day 0. Three control animals underwent necropsy on day 30 post infection.

In total 29 four-week old German landrace pigs were obtained from a commercial high health status herd (BHZP-Basiszuchtbetrieb Garlitz-Langenheide, Germany). This farm is free from the following diseases or pathogens: Pseudorabies, classical swine fever virus (CSFV), Porcine Reproductive and Respiratory Syndrome Virus (PRRSV), *Actinobacillus pleuropneumoniae*, *Mycoplasma hypopneumoniae*, ascaris, mange, *Brachyspira hyodysenteriae* and salmonella category I. Vaccination program does not include vaccination against influenza viruses but the following vaccines were administered: Sows were vaccinated against porcine circovirus type 2 (once), *Erysipelothrix rhusiopathiae*/Porcine parvovirus (twice), salmonella (twice) and *Haemophilus parasuis* (twice). Piglets were vaccinated against PCV2 once and *Mycoplasma hypopneumoniae* twice. Piglets for this study were kept under BSL2 conditions in the animal facilities of the Friedrich-Loeffler-Institut, Greifswald-Riems. IAV infections were performed twice during study period (Table 1), whereby first infection occurred three weeks after transport to our facility by intranasal administration of 2 ml virus suspension ($10^6$ $TCID_{50}$/ml) using mucosal atomization devices (MAD) (Wolfe Tory Medical, USA) with 26 animals. A second infection was performed on day 21 after the first infection. Three animals were mock infected with medium only and served as controls. Blood samples from the same six randomly selected animals were taken on day 0, 2, 4, 7, 14, 21 (prior to second infection), 22, 25 and 31 after first infection for kinetics of blood cells. After euthanasia with Release® (IDT, Germany), necropsy was performed on five animals on day 4, 7, 21 and 25 post first infection. Control animals were euthanized on day 30 after first mock-infection.

## Virus

Influenza virus A/Bayern/74/2009 (hereafter referred to as H1N1pdm09) was propagated on Madin-Darby canine kidney cells (MDCKII) in MEM supplemented with 0.56% bovine serum albumin, 100 U/ml Penicillin, 100 μg/ml Streptomycin and 2 μg/ml L-1-Tosylamide-2-phenylethyl chloromethyl ketone (TPCK)-treated trypsin (Sigma-Aldrich, USA). For viral titration by $TCID_{50}$ assay, serial ten-fold dilutions of virus suspensions were prepared, added to MDCKII cells in 96-well plates, and incubated for three days at 37°C and 5% $CO_2$. Cytopathic effect was examined microscopically. Titers were calculated according to Spearman-Kärber [33, 34]. An acute IAV infection of pigs acquired prior to the study was excluded by real-time PCR (AgPath.ID™ One-Step RT-PCR Kit, Applied Biosystems, USA) of nasal swabs immediately before transport to the experimental facility (modified from [35]).

## Sample preparation

Necropsy and pathological gross examination was performed according to standard guidelines under BSL3** conditions. For extracorporeal bronchoalveolar lavage (BAL), the left main bronchus was cut with sterile scissors and 200 ml of sterile PBS solution was injected with a syringe through the main bronchus into the left lung lobe, which was then kneaded softly. BAL fluid was recovered by syringe.

For flow cytometric analyses organ samples from the following organs were taken during necropsy and stored in ice-cold PBS until further use: mucosa of nasal cavity, lung tissue (*lobus dexter medius*) and lung lymph node (*nodi lymphatici tracheobronchiales inferiores*).

Specimen from the following organs were collected for histopathology and fixed in 4% neutral buffered formaldehyde: nasal cavity with conchae, trachea, lungs (*lobus sinister cranialis pars cranialis and caudalis*, *lobus sinister caudalis*, *lobus dexter cranialis*, *lobus dexter medius*, *lobus dexter accessorius*, *lobus dexter caudalis*), lymph node from the lymphocentrum bronchiale.

Lungs were scored for macroscopically detectable atelectasis (reddish-tan, consolidated, lobular to lobar parenchyma) and scored as follows: (shown as percentage of the total parenchyma analyzed for each pulmonary lobe): 0 = no atelectasis, 1 = mild atelectasis (0–30%), 2 = moderate atelectasis (30–60%), 3 = severe atelectasis (60–100%). The max atelectasis score of all seven scored pulmonary lobes was taken to set up one gross lesion score for each individual pig. For histopathological examination, formaldehyde-fixed tissue samples were embedded in paraffin and cut at 3μm. The sections were mounted on Super-Frost-Plus-Slides (Carl Roth GmbH, Karlsruhe, Germany) and stained with hematoxylin-eosin for light microscopical examination using a Zeiss Axio Scope.A1 microscope equipped with 5x, 10x, 20x, and 40x N-ACHROPLAN objectives (Carl Zeiss Microscopy GmbH, Jena, Germany). All tissue sections were scored blind and investigated for signs and severity levels of inflammation (rhinitis, tracheitis, bronchointerstial pneumonia) as follows: 0 = no inflammation, 1 = mild inflammation, 2 = moderate inflammation, 3 = severe inflammation. The final score for each organ of an individual animal was raised on the basis of the max value of the respective scores.

Immunohistochemistry was performed to detect Influenza A virus antigen in paraffin embedded tissue sections using a mouse monoclonal antibody against the matrixprotein of influenza A virus (M21C64R3, ATCC, Manassas, VA) as previously described [36]. Briefly, tissue sections were dewaxed and rehydrated, and endogenous peroxidase was blocked with 3% $H_2O_2$ (Merck, Darmstadt, Germany) for 10 min. After demasking of antigens with 10mM Na-citrate buffer (pH = 6, 700 W) for 20 min in a microwave oven, sections were incubated with undiluted normal goat serum for 30 min at room temperature to block nonspecific binding sites. Thereafter, section were incubated with anti-Influenza A matrix protein antibodies, diluted 1: 200 in Tris-buffered saline (TBS) at 4°C overnight. This incubation was followed by a biotinylated goat anti-mouse IgG (1:200, LINARIS biologische Produkte, Dossenheim, Germany) and the avidin-biotin-peroxidase complex (Vector Laboratories, Burlingame, CA) for 30 min each at room temperature. Positive antigen-antibody reactions were visualized by incubation with AEC-substrate (Dako, Hamburg, Germany) for 10 min. Sections were washed with deionized water and counterstained with Mayer's hemalaun for 30 s. Sections that stained positive for IAV matrix protein were investigated for cell-specific viral antigen localization and scored for viral antigen distribution in each inspected organ: 0 = negative, 1 = focal to oligofocal, 2 = multifocal, 3 = diffuse as previously described [37]. For each organ the max values of all scores were taken into analysis.

## Cell preparation and antibody staining for flow cytometric analysis

Whole blood was diluted with PBS 1:10 and overall number of white blood cells and different leukocyte subpopulations were determined by blood counting device. Flow cytometric analysis of single cell suspensions from mucosa of nasal cavity, BAL, lung tissue and lung lymph node were performed. Cell from BAL fluid were enriched by centrifugation and discarding the supernatant. Leukocytes from mucosa of nasal cavity, lung lymph node and spleen were prepared by mechanical disruption on metal sieves with plungers and washed with PBS. For

isolation of leukocytes from the lung, tissue was minced with scissors, resuspended in PBS-EDTA supplemented with 100 μM $CaCl_2$, and digested with Collagenase D (1 mg/ml; Sigma-Aldrich) for 40 min at 37˚C. After pressing the digested tissue gently through a cell strainer with a plunger, remaining tissue was removed by short centrifugation.

Cell pellets were suspended in FACS buffer for flow cytometric antibody staining. Antibodies used in this study are listed in S1 Table. Unless otherwise stated, all incubation steps were carried out at 4 ˚C in the dark for 15 min in case of extracellular staining and for 30 min for intracellular staining. Staining of whole blood required erythrocyte lysis after surface staining by conventional lysis buffer (1.55 M $NH_4Cl$, 100 mM $KHCO_3$, 12.7 mM $Na_4EDTA$, pH 7.4, in A. dest.). For intracellular staining, the True-Nuclear™ Transcription Factor Buffer Set was used according to manufacturer's instructions (BioLegend, Germany).

### Software and statistics

Flow cytometric analyses were run on BD FACS CantoII with BD FACS DIVA Software and analyzed using FlowJo Software. GraphPad Prism was used to visualize data and perform statistical analyses. All animal groups examined on days 4, 7, 21, 25 and 30 (control) were analyzed by the nonparametric Kruskal-Wallis test followed by pairwise Dunn's *post hoc* tests compared to control. For blood analyses same tests were used but *post hoc* test was compared to day 0. Statistical significance was designated as $p \leq 0.05$ indicated by an asterisk (*) in the graphs.

## Results

### Intranasal infection of pigs with H1N1pdm09 induced macroscopic and microscopic lesions in the lungs

After intranasal primary IAV infection, multifocal, reddish-tan consolidated areas (pulmonary atelectasis) of different sizes were macroscopically observed in inoculated animals after 4, 7 and 21 days (Fig 1A), mainly in the *lobus dexter medius* and in the *lobus sinister cranialis pars caudalis* (Fig 1B). 4 dpi, the animals reached the highest atelectasis score compared to pigs which were analyzed after 25 dpi (Fig 1C). One control pig showed a minimal, focal atelectasis in the *lobus dexter cranialis*.

Inflammatory changes were detected in the nasal mucosa, trachea and lung. Results from histopathological investigations of nasal mucosa and lungs are summarized in Fig 2. Starting at 4 dpi pigs showed mild, focal, necrotizing rhinitis with loss of epithelial cells (Fig 2 left panel) and IAV matrix protein-positive respiratory epithelial cells within the lesions. Mild, focal, subacute, lymphohistiocytic rhinitis have been observed 7, 21 and 25 dpi. Until 25 dpi inflammation decreased constantly whereas control pigs were free of rhinitis. One infected pig showed mild, necrotizing tracheitis at 4 dpi compared to all other infected and control pigs, which lacked similar lesions. Lung lesions were mainly localized in bronchi, bronchioles and bronchioloalveolar transition zone leading to mild bronchiolointerstitial pneumonia as shown in Fig 2 (right panel). 4 dpi, mild necrosis and loss of bronchial and bronchiolar epithelium was evident in H1N1pdm09 inoculated pigs followed by the infiltration of lymphocytes, macrophages and few neutrophils into the affected tissue (Fig 2C, right panel). At 7 dpi, mild alveolar edema was present whereas necrosis extended to the bronchi-alveolar transition zone (Fig 2E, right panel). At that time, lymphocytes and macrophages increasingly infiltrated the pulmonary interstitium (Fig 2E, right panel), but Influenza A matrix protein was not detectable at any time point later than 4 dpi (Fig 2B, 2D, 2F, 2H and 2J, right panel). 21 dpi, inflammatory cells were still evident (Fig 2G, right panel). Still negative for viral antigen (Fig 2J, right panel), the amount of infiltrating inflammatory cells slightly decreased at 25 dpi (Fig 2I, right panel).

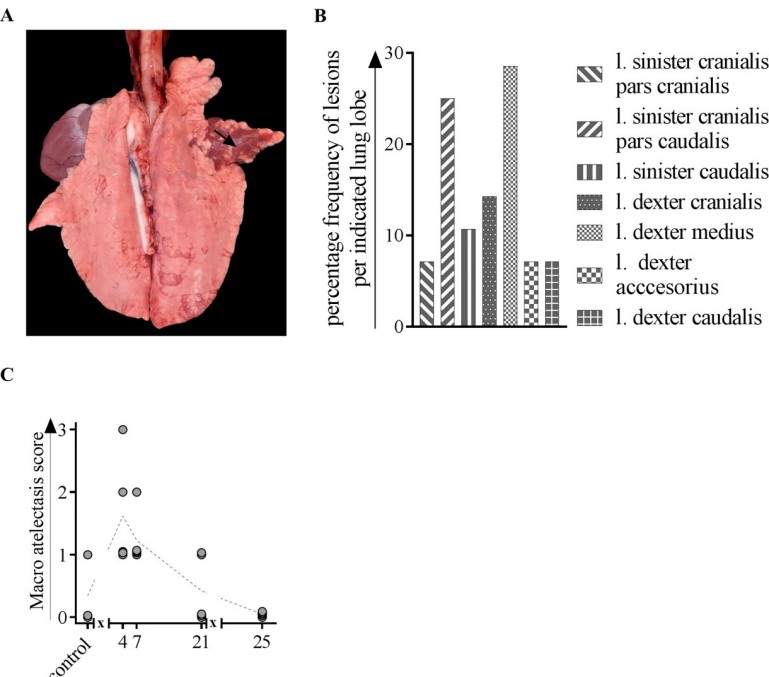

**Fig 1. Gross pathologic changes after macroscopic investigation of lungs from H1N1pdm09-infected pigs.** At indicated time points, three to five animals were subjected to necropsy for macroscopic investigation of artelectasis. (A) Lung from a pig inoculated with H1N1, 4 dpi. Acute, diffuse atelectasis of the *lobus dexter medius* (arrow). (B) Frequency distribution of macroscopic lesions in different lung lobes. (C) Atelectasis scores after 4, 7, 21 and 25 days of H1N1-inoculated and mock-infected animals. l. = lobus; *x* in graph axis indicates infection.

Data from histopathological scoring are summarized in Fig 3. As indicated, IAV matrixprotein was only detectable 4 dpi in the nose, trachea and lung (Fig 3A). At 7 dpi, infected animals showed the highest inflammation score in the nose and lung which then slightly decreased and remained constant until the end of the experiment (Fig 3B). Of note, a moderate significant positive correlation (Spearmann r = 0.464; p<0.0001) was found between macroscopic (atelectasis) and microscopic lesions in the lung (p < 0.0001) (Fig 4).

### IAV H1N1pdm09 infection led to a shift in distribution of blood immune cell subsets but not to leucopenia

Counts of total and different subpopulations of white blood cells did not change significantly during the course of infection (Fig 5). A slight decrease in total cell numbers was observed two days after the first infection but they recovered until day four and stayed at a comparable number until day 14 post first infection. On day 21, prior to second infection, cell number was slightly elevated, stayed at a comparable level the following day and returned to cell numbers comparable to those on day 0 and remained stable until the end of the study (Fig 5A).

Of the myeloid cells in blood, dendritic cell and neutrophil count decreased initially on day two post infection but their numbers as well as monocyte count increased on day four post infection (Fig 5B). On average, monocyte and neutrophil counts remained stable from day 14 post infection until the end of the study, regardless of a second infection, whereas dendritic cell numbers decreased to a hardly detectable level after the second infection and did not recover until the end of the study (Fig 5B). Although neutrophil counts decreased initially after first infection, the frequency of CD14 expressing cells among them increased after first as well as after second infection (Fig 5C).

Regarding cell number of lymphocytes, we observed a decrease for αβ and γδ T cells, as well as for B cells initially after first infection that lasted until day seven and for the latter until the end of the study (Fig 5D). On day 14 post infection cell numbers of γδ T cells and B cells recovered, but αβ T cells numbers were elevated compared to day 0. After second infection αβ T cells decreased to basal level and remained unchanged for the rest of the study, whereas second infection led to another increase in γδ T cells on day 22. Three days later (day 25) and on day 31 numbers of these cells reflected basal levels (Fig 5D).

## Proliferation of CD8⁺ αβ T cells in the blood was increased upon infection with H1N1pdm09

In line with absolute cell numbers obtained from blood counting device (Fig 5), we also observed a decreased frequency of αβ T cells from lymphocytes both after first and second infection with H1N1pdm09 in blood of pigs (Fig 6A). 14 days after first infection, they returned to normal levels. After the second infection recovery time was faster and original frequencies were reconstituted on day 25 (four days after second infection).

To determine subpopulations, frequencies of T cells expressing CD4 and/or CD8 were distinguished into naïve Th cells (CD4⁺/CD8⁻), extrathymic CD4⁺/CD8⁺ cells (including memory as well as cytotoxic cells) and cytolytic T cells (CD4⁻/CD8⁺) (Fig 6B). Overall frequencies of different subpopulations remained stable until day 14 after first infection. Afterwards frequency of cytolytic T cells in bloodstream increased at the expense of naïve cell frequencies. From day 25 (4 days after second infection) increasing frequencies of double-positive Th cells were detected with further decreasing naïve cells. Frequency of proliferating CD8⁺ cells–characterized by Ki-67 expression–increased both after first and to a lesser extent after second infection with H1N1pdm09. CD8⁻ αβ T cells did not show signs of proliferation (Fig 6C).

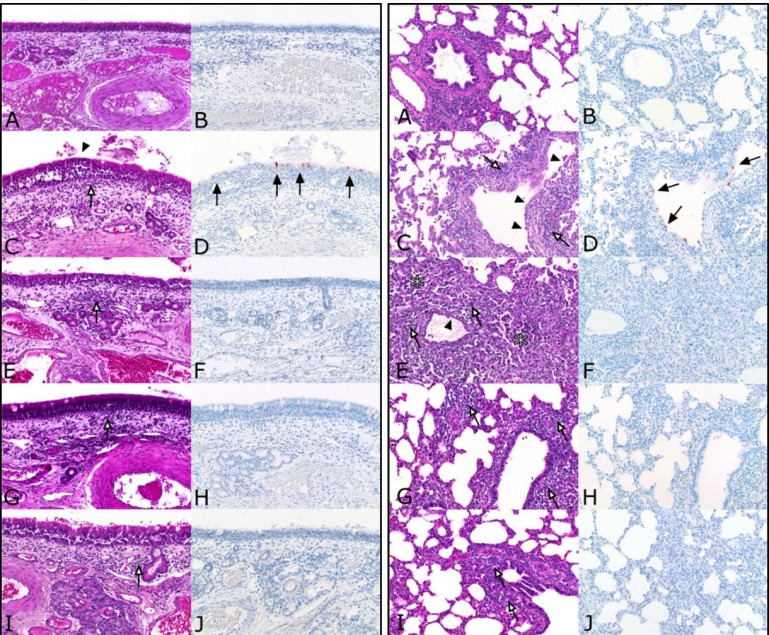

**Fig 2. Histopathology from nose (left panel) and lung (right panel) of H1N1-infected pigs.** At indicated time points, three to five animals were subjected to necropsy. Lungs, trachea and conchae were fixed in 4% formaldehyde, embedded in paraffin and cut at 3μm. Hematoxylin-Eosin (A, C, E, G, I) and anti-Influenza matrixprotein immunohistochemistry (B, D, F, H, J) were performed on nose and lung tissue. A-B) mock-control. (C-D) 4 dpi. (E-F) 7 dpi. (G-H) 21 dpi. (I-J) 25 dpi. White arrows: infiltration of inflammatory cells; black arrows: Influenza A matrix protein-positive cells; arrowheads: flattening and loss of epithelial cells; asterisks: necrotic lung tissue.

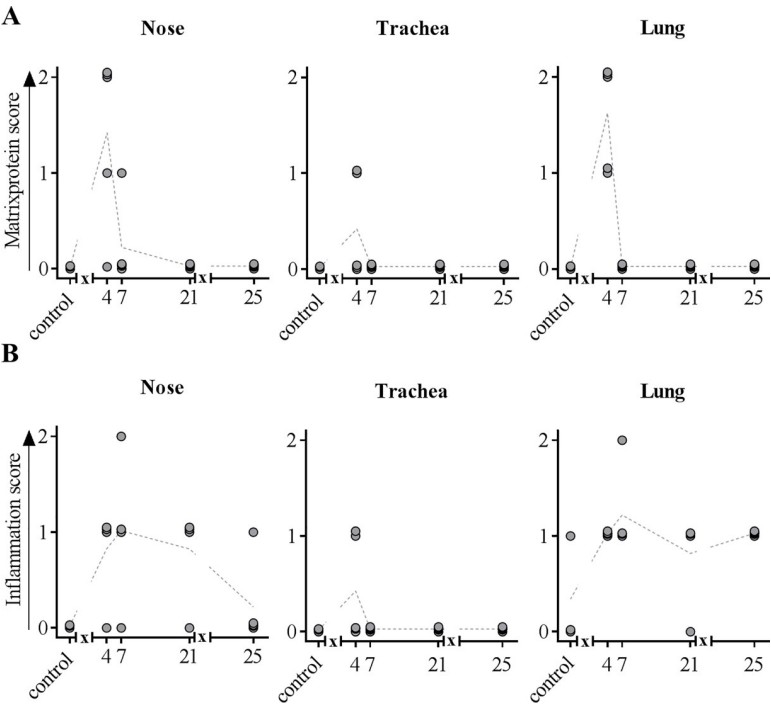

**Fig 3. IAV matrixprotein score (A) and inflammation score (B) obtained from the nose, trachea and lung.** At indicated time points, three to five animals were subjected to necropsy. Lungs, trachea and conchae were fixed in 4% formaldehyde, embedded in paraffin and cut at 3μm. Hematoxylin-eosin staining was used to determine inflammatory score, immunohistochemistry allowed the detection of IAV matrixprotein. *x* in graph axis indicates infection.

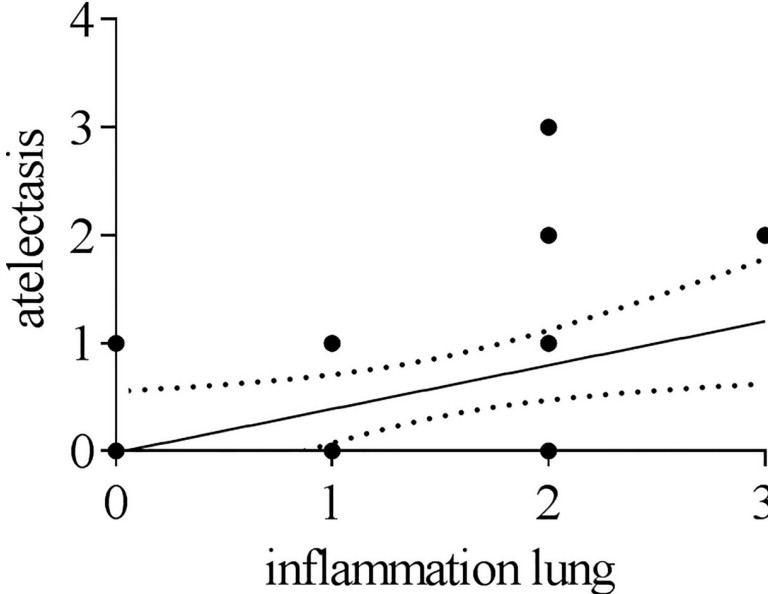

**Fig 4. Correlation between macroscopic and microscopic lesions.** Correlation between scores of atelectasis and simultaneous inflammatory processes in the lung obtained by histopathological investigation. Spearman r = 0,4642 (p < 0,0001).

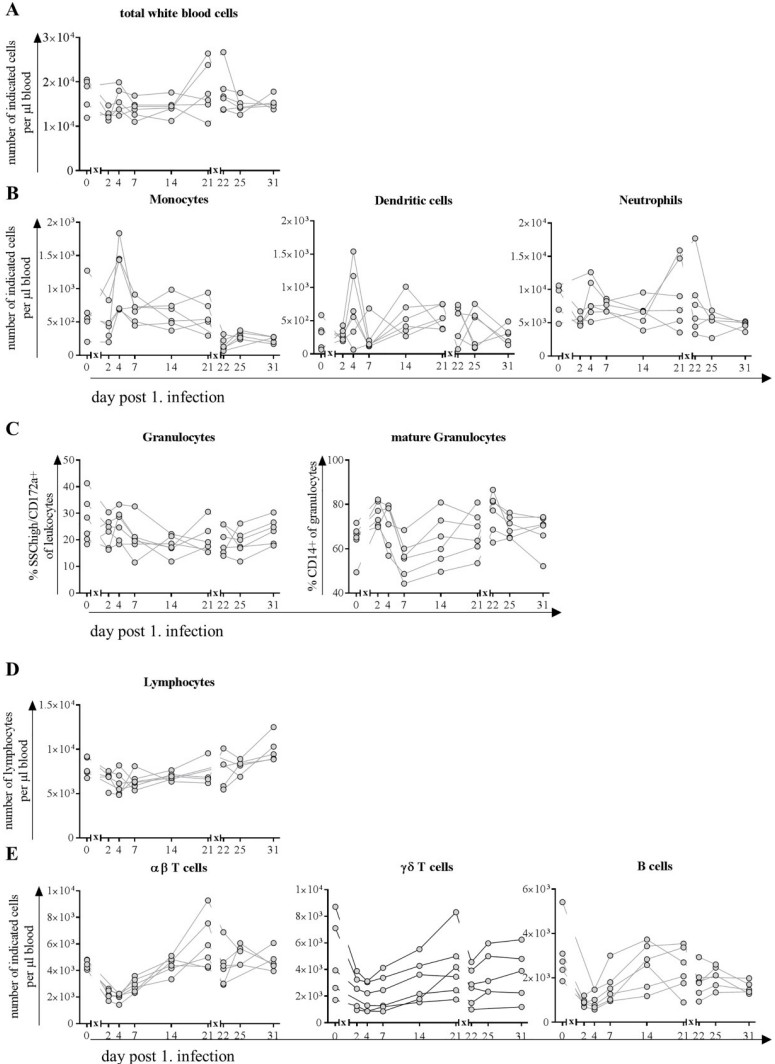

**Fig 5. Counts of peripheral blood leukocytes after infection with H1N1pdm09.** At indicated time points, blood was collected from the same six pigs, randomly chosen on day 0. Blood counting device revealed total count of A) white blood cells and B) myeloid cells (monocytes, dendritic cells and neutrophils). C) Within the granulocytic population frequency of mature granulocytes is indicated by expression of CD14. D) Total count of lymphocytes and E) main subpopulations: αβ T cells, γδ T cells and B cells. *x* in graph axis indicates infection.

## γδ T cells proliferated and increased CD8 expression in the blood after infection with H1N1pdm09

Because γδ T cells do play a major role in pigs during infections, we analyzed these cells and the distribution of different subtypes in the course of IAV infection. Frequencies remained stable during the first infection but were increased immediately after the second infection, which was congruent with decreased frequencies of αβ T cells (Fig 7A).

CD2 and subsequent CD8 expression on the cell surface of γδ T cells resembles activation and maturation steps, the former being observed 14 days after first and one day after second infection with H1N1pdm09 (Fig 7B). Differentiated effector γδ T cells (characterized by simultaneous expression of CD2 and CD8) increased after second infection only. In addition, Ki-67 expressing cells increased among activated and differentiated effector memory T cells after the

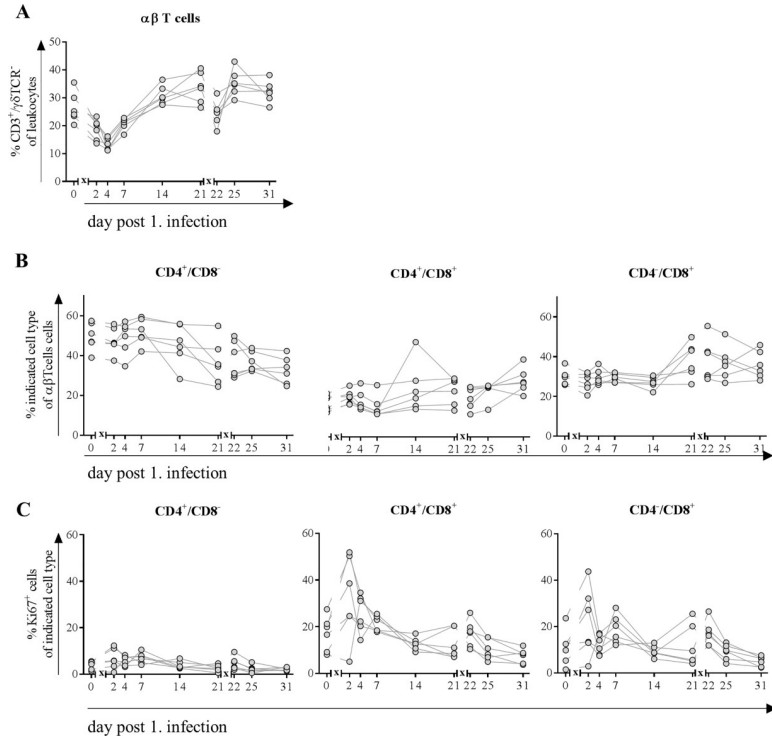

**Fig 6. Frequencies of αβ T cells (A), their subpopulations (B) expressing Ki67 (C).** At indicated time points, blood was collected from the same six pigs, randomly chosen on day 0. Flow cytometric analyses were performed to determine frequency of CD3$^+$/ γδTCR$^-$ T cells (A). Further classification (B) was made based on the expression of CD4$^+$ single positive cells (naive Th cells), CD4$^+$/CD8$^+$ (memory as well as cytotoxic T cells) and CD4$^-$/CD8$^+$ cytolytic T cells. Expression of Ki67 (C) indicated proliferative capacity of cells. *x* in graph axis indicates infection.

first infection (Fig 7C). After the second infection, only the latter showed higher frequencies of proliferating cells. Naïve γδ T cells did not show proliferative activity.

## Frequency of αβ T cells along the respiratory tract increased after the first infection only and expressed mainly CD8αα homodimers

To investigate the immune response along the proposed route of infection, frequencies and functional properties of αβ T cells from lymphocytes in mucosa from nasal cavity, BAL, lung and lung lymph node were analyzed. Increasing frequencies for these cells could be observed four days after first but not after second infection in mucosa of nasal cavity, BAL, lung tissue and lung lymph node and until day seven after first infection in the former two (Fig 8A). Frequencies were still elevated on day 21 compared to control animals and returned to normal levels on day 25, four days after second infection in all organ samples.

Differentiation in subtypes expressing either CD8αα homo- or CD8αβ heterodimers revealed that under physiological conditions (control) CD8$^+$ αβ T cells from tissues of the respiratory tract–nose, BAL and lung–are mainly composed of cells expressing the CD8αα homodimers (Fig 8B). In contrast, in the lymph node CD8αβ expressing T cells were the predominant population. After an initial slight drop in nose and BAL four days after first infection, CD8αα expressing cells increased until day seven in all organs, whereby the ratio in the lymph node was completely reversed towards the homodimer expressing CD8$^+$ T cells. On day 21, prior to second infection, ratio of CD8αα to CD8αβ expressing cells was the same as for control animals. Four days after second infection, the previously observed ratio shift was

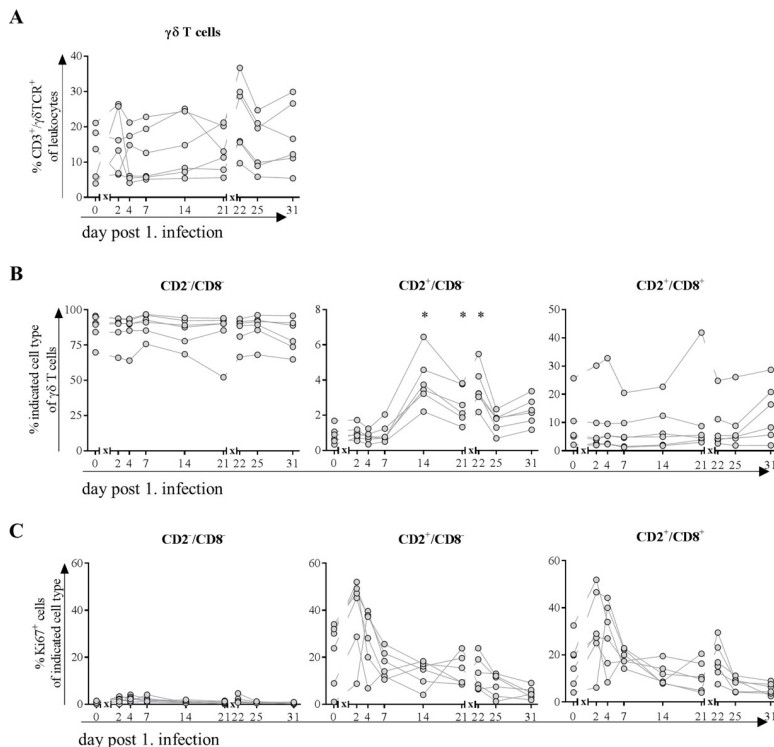

**Fig 7. Frequencies of γδ T cells (A), their subpopulations (B) expressing Ki67 (C).** At indicated time points, blood was collected from the same six pigs, randomly chosen on day 0. Flow cytometric analyses were performed to determine frequency of CD3+/ γδTCR+ T cells. Further classification (B) was made based on the expression of CD2 and CD8: CD2-/CD8- = naïve γδ T cells, CD2+/CD8- = activated γδ T cells and CD2+/CD8+ = differentiated effector γδ T cells. Expression of Ki67 (C) indicated proliferative capacity of cells. * = p≤0.05 Kruskal-Wallis test followed by Dunn's *post hoc* test compared to day 0. *x* in graph axis indicates infection.

repeated and resembled the ratio on day seven except for cells in the lung lymph node, for which the ratio was the same as on day four (Fig 8B).

## In both αβ T cell subsets expression of perforin increased after the first infection

To investigate the functionality of αβ T cells, expression levels of perforin were investigated along the infection route. In all organs, expression of perforin in CD8αα+ cells increased on the fourth day after first infection but returned to levels as observed in control animals already three days later and did not change during the rest of the study (Fig 8C). CD8αβ+ T cells showed a similar pattern, although the level of expression was higher, both under basal (control) as well as in infection conditions (day 4) (Fig 8D). For CD8αβ+ T cells in lung and BAL expression of perforin returned to basal levels on day 7 and remained unchanged. In lung and to a greater extent in the lymph node, expression of perforin was lowest on day 7, returned to basal levels on day 21 but decreased once again after the second infection (Fig 8D).

## Frequency of γδ T cells in the nose and CD8 expressing subpopulations were increased after first infection

Because frequencies of γδ T cells remained stable in the blood, we investigated whether there would be changes in organs from the respiratory tract. In the nose, frequency of cells increased slightly over the fourth to seventh day, stayed elevated until day 21 and returned to levels

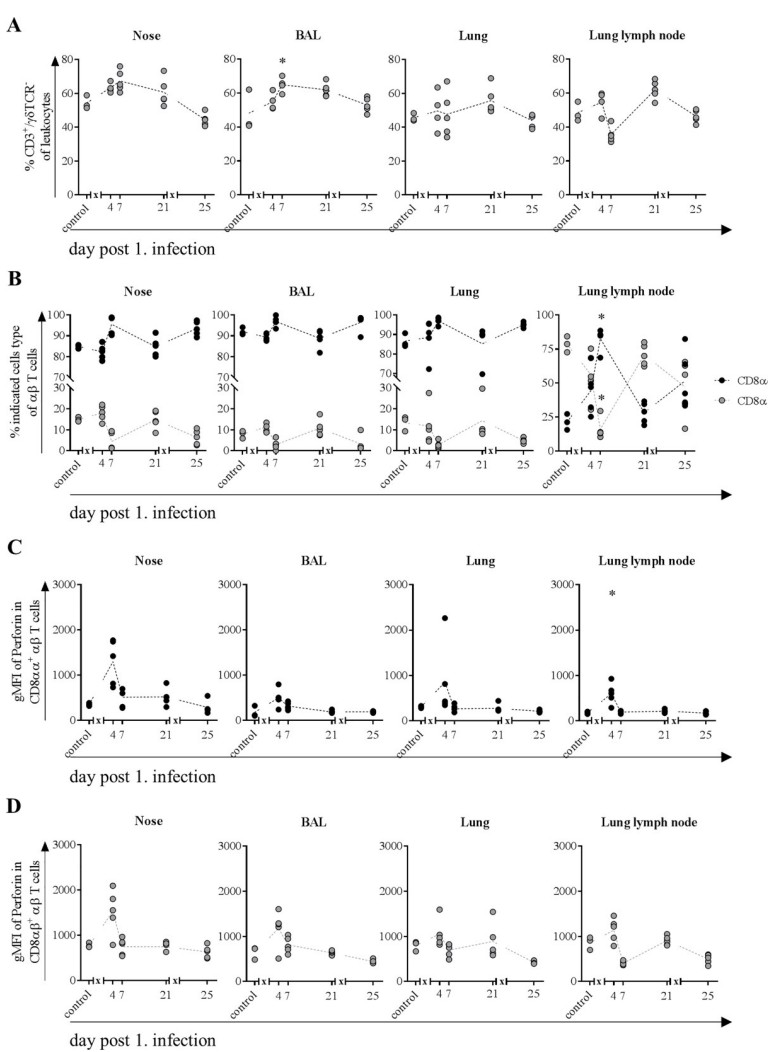

**Fig 8. Frequencies of αβ T cells (A) and CD8αα⁺ and CD8αβ⁺ subpopulations (B) expressing perforin (C and D).**
At indicated time points, three to five animals were subjected to necropsy. After preparation of single cell suspensions from nasal mucosa, BAL, lung tissue and lung lymph node, flow cytometric analyses were performed to determine frequency of CD3⁺/ γδTCR⁻ T cells (A). Further classification (B) was made based on the expression of CD8αα homo- or CD8αβ heterodimers. MFI (mean fluorescence intensity) indicated cytolytic activity of CD8αα (C) and CD8αβ (D) T cells. * = p≤0.05 Kruskal-Wallis test followed by Dunn's *post hoc* test compared to control. *x* in graph axis indicates infection.

comparable to those in control animals on day 25 (four days after second infection) (Fig 9A). In BAL, frequencies continuously decreased over study period. This also applies to the lymph node, albeit to a lesser extent. In the lung, frequency of γδ T cells halved after first and decreased by 1/3 after second infection (Fig 9A).

Investigations on the activation status revealed that in the nose around 40% and in BAL, around 60% of γδ T cells already express the CD8α molecule under physiological conditions (Fig 9B). In contrast to cells in BAL, where frequencies decreased by 10%, activated γδ T cells increased to 60% in the nose four days after infection. In lung and lymph node, frequencies increased from 20% to more than 40% on day four after first infection. For nose, lung and the lymph node, frequency of activated γδ T cells returned to basal levels on day seven and stayed there for every other time point (Fig 9B).

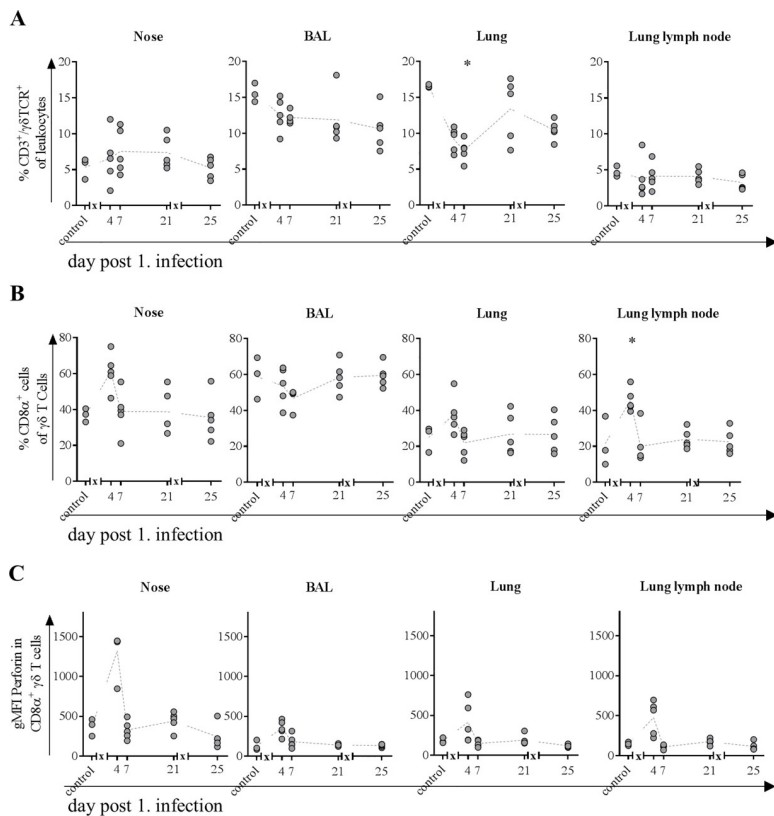

**Fig 9. Frequencies of γδ T cells (A) and CD8α⁺ subpopulation (B) expressing perforin (C).** At indicated time points, three to five animals were subjected to necropsy. After preparation of single cell suspensions from nasal mucosa, BAL, lung tissue and lung lymph node, flow cytometric analyses were performed to determine frequency of CD3⁺/ γδTCR⁺ T cells (A). The CD8α expressing subpopulation (B) was further tested for intracellular perforin expression (C). * = p≤0.05 Kruskal-Wallis test followed by Dunn's *post hoc* test compared to control. *x* in graph axis indicates infection.

## Nasal and BAL γδ T cells show increased expression of perforin and/or transcription factors associated with activation

Related to effector functions of γδ T cells in nose perforin expression was increased only at day four after first infection and did not change at any other day of study period (Fig 9C). For γδ T cells in BAL, lung and lung lymph node, expression of effector molecule perforin was slightly increased on the same day.

Because activation of γδ T cells does not necessarily lead to upregulation of CD8 or perforin, the transcription factors T-bet and EOMES of these cells were also investigated in mucosa from the nasal cavity as well as the BAL. Frequency of γδ T cells expressing T-bet only increased in the nose four days after infection, whereas EOMES single-positive cells increased four days after second infection only (Fig 10A). Frequency of cells expressing both transcription factors did not change during study period. In BAL T-bet expressing cells increased both after first and second infection, whereas EOMES as well as double positive cells remained unchanged (Fig 10B).

## Discussion

Using two different approaches, continuous peripheral blood cell analysis from the same animals as well as analysis of organ samples from the respiratory tract at specific time points after

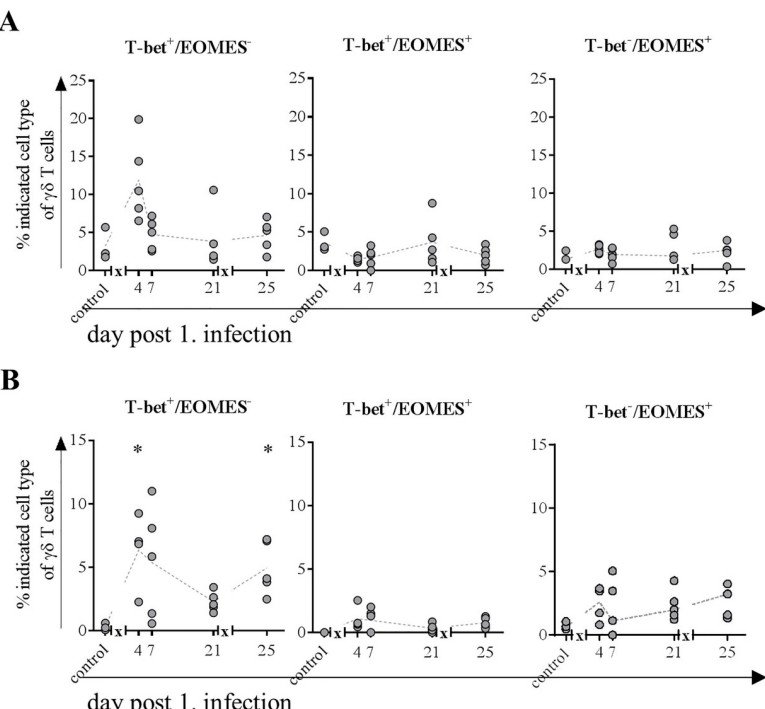

**Fig 10. Frequencies of γδ T cells expressing transcription factors EOMES and/or T-bet.** At indicated time points, three to five animals were subjected to necropsy. After preparation of single cell suspensions from nasal mucosa and BAL, flow cytometric analyses were performed to determine frequency of T-bet⁺/EOMES⁻, T-bet⁺/EOMES⁺ and T-bet⁻/EOMES⁺ γδ T cells in nose (A) and BAL (B). * = p≤0.05 Kruskal-Wallis test followed by Dunn's *post hoc* test compared to control. *x* in graph axis indicates infection.

infection, we characterized in detail the kinetics of the porcine immune response after intranasal infection with H1N1pdm09. In contrast to previous studies reporting fever as well as clinical sings after intranasal infection of pigs with IAV [10, 14, 38, 39], pigs in our study did not develop either. Since different virus strains as well as doses were used, this finding was not unexpected. While a study by Pomorska-Mól *et al.* reported mild clinical signs without fever in intranasally H1N1-infected pigs they neither observed significant changes in hematological parameters nor in numbers of subsets of T and B cells [40]. However, the decreasing lymphocyte count caused by decreasing numbers of T cells and B cells within the first week after infection, correlated well with the other studies [17, 41]. Khatri *et al.* detected also increasing frequencies of CD8⁺ αβ as well as γδ T cells in lungs and lung LN of H1N1-infected pigs [17], which is in line with our findings. We could further show that this also holds true at the site of first contact, the nose. Increased numbers correlated with an increased amount of perforin pointing toward enhanced cytotoxic function. An increase in CD4⁺ T cells in the lungs, which was reported by two groups that observed high fever and strong clinical signs [14, 17], was not observed in our study.

The overall leukocyte count was only slightly decreased on day two after the first infection, which was mainly due to a decrease of the neutrophil population and to a lesser extent to decreasing numbers of αβ T cells and B cells. Two days later, absolute numbers of WBC were restored to normal levels because of elevated numbers of myeloid cells–monocytes, dendritic cells and neutrophils–whereas numbers of lymphocytes decreased further until day seven. These observations parallel the mild lymphopenia with monocytosis in human patients infected with IAV [42–44], which was proposed as a screening tool for influenza infection [45,

46]. An increase in neutrophils of IAV patients within the first days of infection described by some groups [42, 47, 48] could not be detected in our study. However, we observed an increase in frequency of CD14 expressing cells after both infections. CD14 upregulation is associated with activation in granulocytes [49, 50] and serves as a coreceptor for TLR7 mediated recognition of ssRNA [51]. We detected a higher number of dendritic cells in the blood in infected pigs four days after the first infection with simultaneously increased number of monocytes, which was not reported for human patients infected with H1N1pdm09. This might be because in most of the studies characterizing the immune response in H1N1 patients definite time points are hard to determine e.g. time of the infection. In contrast, the decrease in αβ T cells and B cells in pigs covers a period of four to seven days with recovery until day 14 and is comparable with decreasing frequencies in patients infected with H1N1pdm09 although the period of mild lymphopenia varies between studies [43, 46, 52]. However, after the second infection, recovery was faster compared to the first infection and thus comparable to vaccinated humans [53, 54]. Further, we detected an increased proliferative activity in $CD4^+/CD8^+$ T cells (constituting of cytotoxic and memory cells) and cytotoxic ($CD8^+$) subtypes of αβ T cells, which was pronounced after first infection with a lesser increase after second infection resembling a memory response. These observations are in line with the findings in an experimental human influenza experiment that additionally reported those $CD4^+$ T cells to have cytolytic and thus direct antiviral characteristics like perforin expression [55, 56]. Description of prominent $CD4^+$ T cell responses being strong in numbers, by contrast, are primarily associated with severe influenza cases [57].

Given that the H1N1pdm09 infection of the pigs in our setup with MAD leads to an infection that is almost exclusively localized to the respiratory tract, the increase in αβ T cells frequencies in nose and BAL were more prominent starting as early as day 4 post infection and peaking at day 7. Interestingly, in 14 weeks old healthy pigs the ratio of CD8αα to CD8αβ expressing αβ T cells isolated from nasal mucosa, the BAL or lung tissue is shifted towards CD8αα cells with frequencies of 85%, 93% and 85% respectively. This is the first description of the distribution of these two subtypes in respiratory organs of pigs and is in line with finding in humans and mice investigating the role of CD8αα expressing cells among the intraepithelial leukocyte population [58–60]. Because CD8αα is known to be a corepressor of TCR avidity and diminishes activation [61, 62], it is tempting to speculate that infection with H1N1pdm09 leads to activation whilst upregulation of CD8αα inhibits at the same time an excessive immune response. In line with this, due to homeostasis, frequencies of cytolytic CD8αβ T cells decreased further enhancing an anti-inflammatory environment. Several studies in mice and humans do further attribute CD8αα expressing T cells to have memory function [63–65]. Our findings point in the same direction, as CD8αα expression of αβ T cells in the respiratory tract of pigs increased more rapidly after second infection. In control (healthy) pigs, the expression level of perforin per cell within the CD8αα subpopulation in respiratory tract samples is only a third of the expression level in CD8αβ expressing T cells supporting in addition that they do not primarily have cytolytic activity. CD8αα expressing T cells arise mainly from CD8αβ T cells by downregulation of the β-chain and thus diminishing activation [64, 65]. Therefore, it is conceivable that the absence of perforin expression after the second infection is due to inhibition by increased frequencies of CD8αα T cells.

Because lymphocytes in the blood of pigs comprise up to 50% of γδ T cells (~3000 cells/μl blood), which strongly contrasts the maximum of 5% (<500 cells/μl blood) in healthy humans [66], it is challenging to draw conclusions with regard to comparability. Nevertheless, absolute numbers of these cells remained relatively constant throughout the study decreasing only slightly after the first infection but increasing after the second infection at the expense of αβ T cells in the blood. The latter observation might be explained by the different time points of

blood sampling: after the first infection, blood was taken at day two whereas after the second infection it was taken on the first day p.i., to gather a potential memory response. The hypothesis is supported by a pronounced increase in proliferative activity in both activated subtypes of γδ T cells after first infection and a subsequent increase of activated γδ T cells in the blood, supporting that this cell population serves as first line defense. The rapid decrease in frequency of proliferative cells might be explained by the recruitment to the site of inflammation (nose and lung), where their frequencies increased as early as day four after infection with a pronounced perforin expression. Frequencies of γδ T cells in lungs of pigs are comparable to those of humans [67] and they are known not only to maintain pulmonary homeostasis [68] but also to efficiently kill IAV infected epithelial cells and macrophages [69, 70]. In line with these findings from human *in vitro* experiments [70, 71], we observed a notable increase in perforin expression of γδ T cells after first infection in mucosa of nasal cavity constituting an entrance for the virus. Furthermore, concurred increase in frequency of CD8 expressing γδ T cells was observed along the route of entry from nose to lung and lung lymph node, in the latter most probably constituting antigen-presenting cells. Because perforin expression did not increase after the second infection and frequencies of γδ T cells in blood where increased only at day 14 p.i., it is likely that these cells play a major role in recovery from influenza infections as described earlier [29]. Finally, also T-bet expression in γδ T cells increased after both infections in nose and BAL, which is associated with improved recovery from influenza infections [72].

Given that in a natural H1N1 infection in humans the day of infection is not exactly clear, it seems obvious that a day-to-day comparison of patient data with the predefined time points in this study is challenging. Further, for obvious reasons, analyses of lung or respiratory epithelial tissue from patients with mild influenza virus infections are limited. However, transiently decreased numbers of lymphocytes along with increased monocyte counts in the blood seems to be a common feature in mild human and porcine subclinical H1N1pdm09 infections. Further, we could show the presence of and increase in CD8αα expressing αβ as well as CD8α$^+$ γδ T cells in mucosa from respiratory tract after infection in pigs, indicating that these cells have the same dual role as in humans. They do rapidly respond with perforin expression to H1N1pdm09 but simultaneously increase expression of the inhibitory CD8αα molecule to prevent excessive harmful immune responses. Therefore, even though pigs in our study did not show overt clinical signs, underlying immune and pathogenic mechanisms seem to be similar. The results from this study further expand the knowledge of the porcine immune response to pandemic IAV infection and thus, support the use of the pig as a large animal model for human seasonal influenza infections. This offers not only a model for testing the efficacy of new influenza vaccines regarding cellular immune responses but also to expand the model for further investigations of influenza induced pneumonia especially in bacto-viral coinfection scenarios.

## Supporting information

**S1 Table. Antibodies used in flow cytometric analyses.**
(TIF)

## Acknowledgments

The authors thank Regine Kasper, Stefanie Knöfel, Silke Rehbein and Silvia Schuparis for outstanding technical assistance. For excellent care of animals and support during trial, we thank the animal keepers Kerstin Kerstel, Thomas Möritz and Lukas Steinke. Further, we thank Christian Loth and Ralf Redmer for necropsy assistance in singular quality.

## Author Contributions

**Conceptualization:** Theresa Schwaiger, Reiner Ulrich, Ulrike Blohm.

**Data curation:** Theresa Schwaiger, Ulrike Blohm.

**Formal analysis:** Theresa Schwaiger, Julia Sehl, Alexander Schäfer, Ulrike Blohm.

**Funding acquisition:** Thomas C. Mettenleiter, Bernd Köllner, Ulrike Blohm.

**Investigation:** Theresa Schwaiger, Julia Sehl, Reiner Ulrich, Ulrike Blohm.

**Project administration:** Thomas C. Mettenleiter.

**Resources:** Theresa Schwaiger, Julia Sehl, Claudia Karte, Alexander Schäfer, Jane Hühr, Charlotte Schröder, Bernd Köllner, Reiner Ulrich.

**Supervision:** Reiner Ulrich, Ulrike Blohm.

**Validation:** Theresa Schwaiger, Julia Sehl, Ulrike Blohm.

**Visualization:** Theresa Schwaiger, Ulrike Blohm.

**Writing – original draft:** Theresa Schwaiger, Julia Sehl, Ulrike Blohm.

**Writing – review & editing:** Theresa Schwaiger, Claudia Karte, Alexander Schäfer, Jane Hühr, Thomas C. Mettenleiter, Reiner Ulrich, Ulrike Blohm.

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
