## [Decision Letter · Decision Letter 0]

16 Jul 2019

PONE-D-19-15088

Experimental H1N1pdm09 infection in pigs mimics human seasonal influenza infections

PLOS ONE

Dear Dr. Blohm,

Thank you for submitting your manuscript to PLOS ONE. After careful consideration, we feel that it has merit but does not fully meet PLOS ONE’s publication criteria as it currently stands. Therefore, we invite you to submit a revised version of the manuscript that addresses the points raised during the review process.

We would appreciate receiving your revised manuscript by September 15, 2019. To enhance the reproducibility of your results, we recommend that if applicable you deposit your laboratory protocols in protocols.io, where a protocol can be assigned its own identifier (DOI) such that it can be cited independently in the future. For instructions see: http://journals.plos.org/plosone/s/submission-guidelines#loc-laboratory-protocols

We look forward to receiving your revised manuscript.

Kind regards,

Balaji Manicassamy, Ph.D.

Academic Editor

PLOS ONE

Journal Requirements:

Reviewers' comments:

Reviewer's Responses to Questions

**Comments to the Author**

1. Is the manuscript technically sound, and do the data support the conclusions?

Reviewer #1: Partly

Reviewer #2: Yes

Reviewer #3: Yes

2. Has the statistical analysis been performed appropriately and rigorously? 

Reviewer #1: N/A

Reviewer #2: Yes

Reviewer #3: Yes

3. Have the authors made all data underlying the findings in their manuscript fully available?

Reviewer #1: Yes

Reviewer #2: Yes

Reviewer #3: Yes

4. Is the manuscript presented in an intelligible fashion and written in standard English?

Reviewer #1: Yes

Reviewer #2: Yes

Reviewer #3: Yes

5. Review Comments to the Author

Reviewer #1: 1. Please furnish information on the pre-screen done on the pigs. This should include both qPCR and serology (ELISA) for IAV, PRRSV, PPV, PCV 2, TTV. if animals were negative prior to D0, what was their status at the end of the study?

2. How were the mock and challenge groups housed? could comparative counts and FACS data for mocks also be clearly identified and shown along with challenge groups?

Reviewer #2: The manuscript by Schwaiger and colleagues describes the characterization of swine immune cell population changes following an experimental infection with a human H1N1 influenza virus. The results are of interest because swine are a natural host of influenza viruses and can also play a role in the zoonotic transmission of non-human influenza strains into the human population. In addition, (as stated by the authors), because of comparable sizes, immunity and physiology, pigs are a valuable animal model for evaluating immune responses, vaccines, antivirals, etc. In general, the manuscript is well written and provides detailed methods and results that can serve as a guide to other groups. To be fair, the authors should also point out some important differences between pigs and humans such as the much shorter “childhood” and general lifespan of pigs; important differences in immune cell populations such as the double positive CD4+CD8+ lymphocytes and strikingly different proportions of gamma-delta+ T cells in blood, or the lack of reagents and general knowledge of the pig’s immunity in comparison to other animal models (mice).

The authors should also consider the following specific comments:

Line 68. The animals were obtained from a commercial herd. If known, it would be important to state the serological status (from natural infection and from vaccination) of the mothers, since the presence of maternal antibodies against H1N1 influenza would have a strong impact in the evolution of the infection.

Line 73. Animals were re-infected with the same influenza strain 21 days after the first infection. In the context of modelling influenza in humans, it is not clear what was the purpose of this infection. In humans, re-infection occurs in the following influenza seasons (at least 1 year later) and by an antigenically drifted strain.

Line 122. “…antibody against the anti-matrixprotein…” this sentence is incorrect. It should be either “…antibody against the matrixprotein…” or “…anti-matrixprotein monoclonal antibody…”. Please correct.

Line 128. It might be useful to describe what TBS stands for.

Line 149. In this paragraph there is a sentence repeated “Antibodies used in this study are listed in Supplementary table 1”. Please correct.

Line 175 and beyond. Please consider using the term “necropsy” instead of “autopsy”.

Line 197. I believe the panels from figure 2 are cited wrongly: panel 2I shows H&E staining (infiltrating inflammatory cells) and panel 2J shows immunohistochemistry (viral antigen).

Line 334. “…as well as in infectious conditions” I believe it should say “…as well as in infection conditions”.

Line 404. This is probably a typo, “deceased” should say “decreased”

Line 423. “…reported those CD4+ T cells to have cytolytic…” please double check. Did you mean CD8+?

Reviewer #3: The manuscript by Schwaiger and colleagues described experimental H1N1pdm09 infection in pigs in order to mimic human seasonal influenza infections. The authors used 2009 pandemic H1N1 virus to intranasally infect a group of 4-week-old pigs for 21 days, then re-infected them with the same virus again to investigate systemic and local immune responses by testing blood, mucosa of nasal cavity and lung tissues bronchoalveolar lavage fluid samples. Result showed that decreasing numbers of peripheral blood lymphocytes were detected after the first infection, the simultaneous increase in the frequencies of proliferating cells correlated with an increase in infiltrating leukocytes in the lung. Furthermore, a cytotoxic T cell response was detected but restricted to the respiratory route of virus entry such as the nose, the lung and the bronchoalveolar lavage according to detected enhanced perforin expression in αβ and γδ T cells in the respiratory tract. Increasing frequencies of CD8αα expressing αβ T cells were also observed rapidly after the first infection, which could play a role in inhibition of uncontrolled inflammation in the respiratory tract. Authors conclude that the results from this study demonstrate that experimental influenza A virus infection in pigs mimics major characteristics of human seasonal influenza A virus infections. The manuscript is well written and provides interesting information which complements findings of former studies.

Comments:

The pigs used in the study were purchased from a commercial high health status herd. The detailed information on these pigs should be provided such as which pathogens (influenza A, D virus, PRRSV, PCV2 and mycoplasma etc) were tested prior to infection as any prior infection will impact the obtained results and further interpret.

In the figures 5, 6, 9 and 10, the data from only infected pigs were presented despite of the data at d0 as controls. If the pigs were infected other pathogens before, these data could not reflect the reality of only infection of 2009 pandemic H1N1 virus. Therefore, it is necessary to provide more detailed information on pigs

6. PLOS authors have the option to publish the peer review history of their article (what does this mean?). If published, this will include your full peer review and any attached files.

Reviewer #1: No

Reviewer #2: No

Reviewer #3: No

---

## [Author Response · Author response to Decision Letter 0]

9 Sep 2019

Journal Requirements:

1. When submitting your revision, we need you to address these additional requirements. Please ensure that your manuscript meets PLOS ONE's style requirements, including those for file naming. The PLOS ONE style templates can be found at http://www.journals.plos.org/plosone/s/file?id=wjVg/PLOSOne_formatting_sample_main_body.pdf

and http://www.journals.plos.org/plosone/s/file?id=ba62/PLOSOne_formatting_sample_title_authors_affiliations.pdf

We thank for this hint from the editors and the links. We checked the manuscript carefully again and changed the style at the required points according to PLOS ONE style requirements.

We thank the editor for this note. Because the data are not a core part of our research, we removed the phrase “data not shown” from line 195. 

Reviewers' comments:

Reviewer #1:

1. Please furnish information on the pre-screen done on the pigs. This should include both qPCR and serology (ELISA) for IAV, PRRSV, PPV, PCV 2, and TTV. If animals were negative prior to d0, what was their status at the end of the study?

Prior to d0, pigs were tested negative for IAV genome by qPCR. During the trial, we were able to detect IAV only on day 2 and 4 after first infection in nasal swabs (see Table 1). Neutralizing antibodies were detected in sera of infected pigs only from day 14 with very low titres until the end of the study (Table 2). Control pigs did not show antibodies against H1N1. 

Table 1 Viral titres during trial in nasal swabs. Nasal swabs were taken throughout the study period and tested for replicable virus by TCID50 assay. Results are given as TCID50/ml supernatant. First three rows are mock-infected control animals (grey filling). neg = negative, nt = not tested. 

Animal d0 d2 d4 d7 d22 d23 d25 Necropsy at day

7936 neg neg neg neg neg neg neg 30

7444 neg neg neg neg neg neg neg 

7460 neg neg neg neg neg neg neg 

7477 neg neg 1,5x 102 nt nt nt nt 4

7470 neg neg 3,2x 103 nt nt nt nt 

7964 neg neg 3,2x 103 nt nt nt nt 

7993 neg 6,8x 103 1,5x 104 nt nt nt nt 

7445 neg 3,2x 103 6,8x 104 nt nt nt nt 

7370 neg 3,2x 102 6,8x 103 neg nt nt nt 7

7410 neg 3,2x 102 6,7x 101 neg nt nt nt 

7468 neg 6,8x 102 1,4x 103 neg nt nt nt 

7479 neg 1,4x 103 3,2x 103 neg nt nt nt 

7965 neg 1,4x 102 1,5x 102 neg nt nt nt 

7403 neg 1,4x 103 7,1x 102 neg nt nt nt 21

7970 neg neg 6,8x 103 neg nt nt nt 

7478 neg 1,4x 104 1,5x 105 neg nt nt nt 

7939 neg neg 6,7x 101 neg nt nt nt 

7434 neg neg 1,5x 104 neg nt nt nt 

7443 neg 6,8x 102 6,8x 102 neg neg neg neg 25

7367 neg 1,4x 102 6,8x 102 neg neg neg neg 

7402 neg 1,5x 104 1,5x 104 neg neg neg neg 

7480 neg 1,4x 102 3,2x 104 neg neg neg neg 

7471 neg 1,4x 102 6,8x 104 neg neg neg neg 

7407 neg 6,8x 102 1,5x 104 neg neg neg neg 31

7408 neg neg 1,5x 103 neg neg neg neg 

7995 neg neg 1,5x 103 neg neg neg neg 

7991 neg neg 3,2x 103 neg neg neg neg 

7441 neg neg neg neg neg neg neg 

7433 neg neg 1,5x 103 neg neg neg neg 

Table 2. Titers of neutralizing antibodies. During trial serum of naïve (rows 1-3, gray filling) and infected (rows 4-9) was tested by Hemagglutination inhibition test. neg = negative, nt = not tested.

Animal d7 d14 d21 d22 d25 d31

7936 nt nt nt nt nt neg

7444 nt nt nt nt nt neg

7460 nt nt nt nt nt neg

7408 neg 1:160 1:160 1:320 1:320 1:320

7407 neg 1:160 1:160 1:160 1:320 1:160

7991 neg 1:160 neg neg 1:160 neg

7995 neg 1:160 neg neg neg neg

7441 neg 1:320 1:160 1:160 1:160 1:160

7433 neg neg 1:160 1:160 1:320 1:320

We have not yet provided this information, as we do not consider it a core part of this study, which focuses on the cellular mucosal immune response. If desired, we can include viral titres and titration of neutralizing antibodies in new supplementary tables to provide additional information. 

The farm from which the pigs were purchased is free of PRRSV, which obviates the need for re-testing by our laboratory. Furthermore, the sows were vaccinated against PPV on the 170th and 190th day of life and against PCV 2 on day 160. Because we started our study when the animals were 8 weeks old, the maternal antibodies against these viruses were probably degraded as far as possible [1]. Piglets themselves received vaccines against PCV2 and H. parasuis at their 14th day of life, which makes it likely that they already started to generate antibodies against these viruses, which we did not screen for. For the occurrence of TTV was tested neither by us nor by the farm. We included a section on pathogen status and vaccination program of the farm in Material and Methods section (lines 81-89): “This farm is free from the following diseases or pathogens: Pseudorabies, classical swine fever virus (CSFV), Porcine Reproductive and Respiratory Syndrome Virus (PRRSV), Actinobacillus pleuropneumoniae, Mycoplasma hypopneumoniae, ascaris, mange, Brachyspira hyodysenteriae and salmonella category I. Vaccination program does not include vaccination against influenza viruses but the following vaccines were administered: Sows were vaccinated against porcine circovirus type 2 (once), Erysipelothrix rhusiopathiae/Porcine parvovirus (twice), salmonella (twice) and Haemophilus parasuis (twice). Piglets were vaccinated against PCV2 once and Mycoplasma hypopneumoniae twice.”

2. How were the mock and challenge groups housed? 

After arrival of the animals (4-weeks of age), the piglets were randomly separated in three groups and housed in three different sheds under BSL2 conditions. One week prior to infection (7-weeks of age) pigs were divided in six groups, with the control group held in a shed at the other end of the building. Five groups were infected one week later. Contact between the animals of different sheds was strictly avoided not only structurally but also by different hygiene interventions (showers between the sheds, disposable clothing and masks within the sheds).

3. Could comparative counts and FACS data for mocks also be clearly identified and shown along with challenge groups?

With this question, the reviewer addresses one of the central points of our study. The characteristic feature of this study is that the animals intended for organ removal (four infected groups plus control group) were only subjected to blood sample drawing on the day of necropsy. This also includes the control group whose section day coincided with the last day of the study. We deliberately chose this scheme because regular blood collection has a significant effect on the stress level of the animals and thus on the immune response. In order to avoid this effect, the same six pigs randomly selected at the beginning of the experiment were used for the blood kinetics only. Blood from control animals was only drawn at day of their euthanasia. This is why we cannot provide comparative counts and FACS data for peripheral blood leukocytes of mock animals from time points correlating with those of infected blood-donating group. From our point of view, the d0 sample of the latter animals represents the adequate control to all blood samples of these animals after infection.

Reviewer #2:

The manuscript by Schwaiger and colleagues describes the characterization of swine immune cell population changes following an experimental infection with a human H1N1 influenza virus. The results are of interest because swine are a natural host of influenza viruses and can also play a role in the zoonotic transmission of non-human influenza strains into the human population. In addition, (as stated by the authors), because of comparable sizes, immunity and physiology, pigs are a valuable animal model for evaluating immune responses, vaccines, antivirals, etc. In general, the manuscript is well written and provides detailed methods and results that can serve as a guide to other groups.

1. To be fair, the authors should also point out some important differences between pigs and humans such as the much shorter “childhood” and general lifespan of pigs; important differences in immune cell populations such as the double positive CD4+CD8+ lymphocytes and strikingly different proportions of gamma-delta+ T cells in blood, or the lack of reagents and general knowledge of the pig’s immunity in comparison to other animal models (mice).

The reviewer is right. We have added information about drawbacks when using the pig as a model compared to the mouse model (lines 33-37 and lines 44-45): “Studies on the immune response in pigs have long been biased by the lack of specific reagents. Especially the variety of antibodies available for the pig is still far lower to that of mice. As a result, the knowledge of the porcine immune response in general is much smaller compared to that of mice. Despite these disadvantages, pigs have decisive advantages as a model for human IAV infection.”, and “[…] and by expanding the number of reagents for porcine immunological analyses, […]”. We also included a paragraph on the major differences in peripheral blood cell composition between human and swine (lines 57-63 and references 26-32): “However, it is important to note that the prominent porcine populations of CD4+/CD8+ double positive T cells [26] as well as the high number of peripheral γδ T [27] cells are virtually absent in humans [28, 29], representing a major difference. Besides the numerical difference, the functionality of the two cell populations is comparable in both human and swine. CD4+/CD8+ double positive T cells are mature effector cells with memory characteristics that rapidly mount antigen-specific responses upon antigenic challenge [18, 30]. Besides acting as innate immune cells via pattern recognition receptors and direct killing of infected cells, γδ T cells do play a major role in antigen processing and presentation in human and swine [31, 32].”

With regard to the shorter "childhood" of pigs, we cannot agree: compared to the life span of pigs, the "childhood" is not much shorter. Especially regarding the risk and the frequency of infections of the young animals in the flat deck area, the herd veterinarians report amazing parallels to children who come to kindergarten.

2. The authors should also consider the following specific comments:

a. Line 68. The animals were obtained from a commercial herd. If known, it would be important to state the serological status (from natural infection and from vaccination) of the mothers, since the presence of maternal antibodies against H1N1 influenza would have a strong impact in the evolution of the infection.

The reviewer is right, that maternal antibodies have a strong impact in the evolution of infection. In our study, serological status of sows regarding antibodies directed against H1N1 viruses was not taken into account, because our study started, when pigs were 8 weeks of age. By this time, all maternal antibodies should have been largely metabolized [1].

b. Line 73. Animals were re-infected with the same influenza strain 21 days after the first infection. In the context of modelling influenza in humans, it is not clear what was the purpose of this infection. In humans, re-infection occurs in the following influenza seasons (at least 1 year later) and by an antigenically drifted strain.

The reviewer is right that re-infection in humans occurs primarily around a year later with an antigenically drifted strain in the vast majority of cases. The purpose of the re-infection in our study three weeks after first infection was to analyse a potential memory response. This re-infection was not intended to imitate the seasonal flu that occurs in humans a year later. The period of three weeks between the first and second infection in our study allows at most a comparison with the seasonal infection after vaccination in humans.

c. Line 122. “…antibody against the anti-matrixprotein…” this sentence is incorrect. It should be either “…antibody against the matrixprotein…” or “…anti-matrixprotein monoclonal antibody…”. Please correct.

We corrected the sentence as follows: “…antibody against the matrixprotein of influenza A virus…” (line 139).

d. Line 128. It might be useful to describe what TBS stands for.

We added the full form “Tris-buffered saline (TBS)” in the material and methods section (line 145).

e. Line 149. In this paragraph there is a sentence repeated “Antibodies used in this study are listed in Supplementary table 1”. Please correct.

We have deleted one of the duplicate sentences (line 169).

f. Line 175 and beyond. Please consider using the term “necropsy” instead of “autopsy”.

We exchanged the term “autopsy” to “necropsy” throughout the document (lines 190, 221, 229, 325, 365, 392 and 520).

g. Line 197. I believe the panels from figure 2 are cited wrongly: panel 2I shows H&E staining (infiltrating inflammatory cells) and panel 2J shows immunohistochemistry (viral antigen).

We fixed the wrong citation and used the correct labelling to the panels of figure 2 in the text. The new text is as follows: “Still negative for viral antigen (Fig 2J, right panel), the amount of infiltrating inflammatory cells slightly decreased at 25 dpi (Fig 2I, right panel). “ (lines 212-213).

h. Line 334. “…as well as in infectious conditions” I believe it should say “…as well as in infection conditions”.

We corrected the expression “…infectious conditions”. It now reads as follows: “…infection conditions” (line 349).

i. Line 404. This is probably a typo, “deceased” should say “decreased”

We have corrected the typo. The word shall read as follows: “decreased” (line 418).

j. Line 423. “…reported those CD4+ T cells to have cytolytic…” please double check. Did you mean CD8+?

Although it may sound unusual at first, Wilkinson and colleagues showed that influenza-specific CD4+ T cells were able to kill autologous cells in a peptide specific manner [2]. This finding was later confirmed in a study performed by Zhou et al. [3]. Growing amount of evidence suggest CD4+ T cells to have direct antiviral activity and thus, to play a critical role in numerous viral infections in humans and animal models in vivo. These findings are summarized in e.g. these reviews: [4-6].

Reviewer #3: 

The manuscript by Schwaiger and colleagues described experimental H1N1pdm09 infection in pigs in order to mimic human seasonal influenza infections. The authors used 2009 pandemic H1N1 virus to intranasally infect a group of 4-week-old pigs for 21 days, then re-infected them with the same virus again to investigate systemic and local immune responses by testing blood, mucosa of nasal cavity and lung tissues bronchoalveolar lavage fluid samples. Result showed that decreasing numbers of peripheral blood lymphocytes were detected after the first infection, the simultaneous increase in the frequencies of proliferating cells correlated with an increase in infiltrating leukocytes in the lung. Furthermore, a cytotoxic T cell response was detected but restricted to the respiratory route of virus entry such as the nose, the lung and the bronchoalveolar lavage according to detected enhanced perforin expression in αβ and γδ T cells in the respiratory tract. Increasing frequencies of CD8αα expressing αβ T cells were also observed rapidly after the first infection, which could play a role in inhibition of uncontrolled inflammation in the respiratory tract. Authors conclude that the results from this study demonstrate that experimental influenza A virus infection in pigs mimics major characteristics of human seasonal influenza A virus infections. The manuscript is well written and provides interesting information, which complements findings of former studies.

1. The pigs used in the study were purchased from a commercial high health status herd. The detailed information on these pigs should be provided such as which pathogens (influenza A, D virus, PRRSV, PCV2 and mycoplasma etc) were tested prior to infection as any prior infection will impact the obtained results and further interpret.

Piglets were tested negative by qPCR for influenza A virus genome. Neutralizing antibodies were detected in sera of infected pigs only from day 14 post infection until the end of the study. We have not yet provided this information, as we do not consider it a core part of this study, which focuses on the cellular mucosal immune response. If desired, we can include viral titres and titration of neutralizing antibodies in new supplementary tables to provide additional information. 

The farm from which the pigs were purchased is free from PRRSV, Pseudorabies, classical swine fever virus (CSFV), Porcine Reproductive and Respiratory Syndrome Virus (PRRSV), Actinobacillus pleuropneumoniae, Mycoplasma hypopneumoniae, ascaris, mange, Brachyspira hyodysenteriae and salmonella category I. which obviates the need for re-testing by our laboratory. Furthermore, the sows were vaccinated against PPV on the 170th and 190th day of life and against PCV 2 on day 160. Additionally, piglets themselves received vaccines against PCV2 and H. parasuis at their 14th day of life. This vaccination program together with the strict hygiene management (no contact to other animals, showers between the sheds, disposable clothing and masks within the sheds) makes a prior infection rather unlikely.

To address this issue, we included a section on pathogen status and vaccination program of the farm in Material and Methods section (lines 81-89): “This farm is free from the following diseases or pathogens: pseudorabies, classical swine fever virus (CSFV), Porcine Reproductive and Respiratory Syndrome Virus (PRRSV), Actinobacillus pleuropneumoniae, Mycoplasma hypopneumoniae, ascaris, mange, Brachyspira hyodysenteriae and salmonella category I. Vaccination program does not include vaccination against influenza viruses but the following vaccines were administered: Sows were vaccinated against porcine circovirus type 2 (once), Erysipelothrix rhusiopathiae/Porcine parvovirus (twice), salmonella (twice) and Haemophilus parasuis (twice). Piglets were vaccinated against PCV2 once and Mycoplasma hypopneumoniae twice.”

2. In the figures 5, 6, 9 and 10, the data from only infected pigs were presented despite of the data at d0 as controls. If the pigs were infected other pathogens before, these data could not reflect the reality of only infection of 2009 pandemic H1N1 virus. Therefore, it is necessary to provide more detailed information on pigs.

Yes, the reviewer is right, in Fig. 8, 9 and 10 we compared animals at different times of infection with only one control group at one time point. We have opted for this method to avoid excessive animal numbers. It is true that the control animals could also develop infections during the experimental period, so the control group was subjected to necropsy at the end of the experiment and not at the beginning. The animals all come from the same delivery, have the same age and were kept under comparable conditions (see reviewer 1, comment #2), so the error is considered to be minor. We have included information about health status of the pigs and the vaccination regime of the breeding farm in the manuscript. Thank you very much for this note.

The choice of correct controls is always difficult and controversial in these experiments: is the individual animal control (d0 before infection) superior to an untreated control group or vice versa? Animal-individual control has the error that an incalculable handling error (capture, frequent bleeding) is to be considered. In order best to present both situations equally well, we conducted two independent experiments in parallel with animals of the same origin and age: animal-individual non-lethal data from blood (Fig 5-7) and data for the organs after necropsy with an uninfected comparison group (Fig 8-10). 

As stated above (reviewer 3, comment #1), pigs were tested negative for IAV genome by qPCR prior to infection. Additionally the vaccine program of the farm from which we purchased the piglets almost eliminates a prior infection with the mentioned porcine pathogens. Because contact to other animals was strictly avoided and hygiene management (also see reviewer 1, comment #2) was rigorously followed, it is rather unlikely that pigs were infected with other pathogens in the time of acclimatization in our animal facility under BSL2 conditions (four weeks prior to start of experiment). In addition, necropsies did only reveal a single lung lesion in one animal (in blood donating group) that could not be attributed to IAV infection. As already stated above, we included a section in Material and Methods section on pathogen status and vaccination program of the farm (lines 81-89).

References:

1. Reeth KV, Labarque G, Pensaert M. Serological Profiles after Consecutive Experimental Infections of Pigs with European H1N1, H3N2, and H1N2 Swine Influenza Viruses. Viral Immunology. 2006;19(3):373-82. doi: 10.1089/vim.2006.19.373.

2. Wilkinson TM, Li CKF, Chui CSC, Huang AKY, Perkins M, Liebner JC, et al. Preexisting influenza-specific CD4+ T cells correlate with disease protection against influenza challenge in humans. Nature Medicine. 2012;18:274. doi: 10.1038/nm.2612

https://www.nature.com/articles/nm.2612#supplementary-information.

3. Zhou X, McElhaney JE. Age-related changes in memory and effector T cells responding to influenza A/H3N2 and pandemic A/H1N1 strains in humans. Vaccine. 2011;29(11):2169-77. doi: 10.1016/j.vaccine.2010.12.029. PubMed PMID: 21353149.

4. Juno JA, van Bockel D, Kent SJ, Kelleher AD, Zaunders JJ, Munier CM. Cytotoxic CD4 T Cells-Friend or Foe during Viral Infection? Front Immunol. 2017;8:19. Epub 2017/02/09. doi: 10.3389/fimmu.2017.00019. PubMed PMID: 28167943; PubMed Central PMCID: PMCPMC5253382.

5. Soghoian DZ, Streeck H. Cytolytic CD4(+) T cells in viral immunity. Expert Rev Vaccines. 2010;9(12):1453-63. doi: 10.1586/erv.10.132. PubMed PMID: 21105780.

6. Brown DM, Lampe AT, Workman AM. The Differentiation and Protective Function of Cytolytic CD4 T Cells in Influenza Infection. Frontiers in immunology. 2016;7:93-. doi: 10.3389/fimmu.2016.00093. PubMed PMID: 27014272.

---

## [Editor Report · Decision Letter 1]

11 Sep 2019

Experimental H1N1pdm09 infection in pigs mimics human seasonal influenza infections

PONE-D-19-15088R1

Dear Dr. Blohm,

We are pleased to inform you that your manuscript has been judged scientifically suitable for publication and will be formally accepted for publication once it complies with all outstanding technical requirements.

With kind regards,

Balaji Manicassamy, Ph.D.

Academic Editor

PLOS ONE
---

## [Editor Report · Acceptance letter]

13 Sep 2019

PONE-D-19-15088R1 

Experimental H1N1pdm09 infection in pigs mimics human seasonal influenza infections 

Dear Dr. Blohm:

I am pleased to inform you that your manuscript has been deemed suitable for publication in PLOS ONE. Congratulations! Your manuscript is now with our production department. 

With kind regards,

on behalf of

Dr. Balaji Manicassamy 

Academic Editor

PLOS ONE